# Preparation of Hydrogel/Silver Nanohybrids Mediated by Tunable-Size Silver Nanoparticles for Potential Antibacterial Applications

**DOI:** 10.3390/polym11040716

**Published:** 2019-04-19

**Authors:** Yeray A. Rodríguez Nuñez, Ricardo I. Castro, Felipe A. Arenas, Zoraya E. López-Cabaña, Gustavo Carreño, Verónica Carrasco-Sánchez, Adolfo Marican, Jorge Villaseñor, Esteban Vargas, Leonardo S. Santos, Esteban F. Durán-Lara

**Affiliations:** 1Instituto de Química de Recursos Naturales, Universidad de Talca, Talca 3460000, Maule, Chile; 2Multidisciplinary Agroindustry Research Laboratory, Carrera de Ingeniería en Construcción e Instituto de Ciencias Químicas Aplicadas, Universidad Autónoma de Chile, Talca 3460000, Chile; 3Departamento de Biología, Facultad de Química y Biología, Universidad de Santiago de Chile, Santiago 8320000, Chile; 4Bio & NanoMaterials Lab| Drug Delivery and Controlled Release, Universidad de Talca, Talca 3460000, Maule, Chile; 5Departamento de Microbiología, Facultad de Ciencias de la Salud, Universidad de Talca, Talca 3460000, Maule, Chile; 6Center for the Development of Nanoscience and Nanotechnology, Santiago 8320000, Chile

**Keywords:** silver nanoparticles, hydrogel, crosslinking, template, maleic acids, polyvinyl alcohol, antibacterial activity

## Abstract

In this study, a versatile synthesis of silver nanoparticles of well-defined size by using hydrogels as a template and stabilizer of nanoparticle size is reported. The prepared hydrogels are based on polyvinyl alcohol and maleic acid as crosslinker agents. Three hydrogels with the same nature were synthesized, however, the crosslinking degree was varied. The silver nanoparticles were synthesized into each prepared hydrogel matrix achieving three significant, different-sized nanoparticles that were spherical in shape with a narrow size distribution. It is likely that the polymer network stabilized the nanoparticles. It was determined that the hydrogel network structure can control the size and shape of the nanoparticles. The hydrogel/silver nanohybrids were characterized by swelling degree, Thermal Gravimetric Analysis (TGA), Fourier Transform Infrared (FT-IR), Scanning Electron Microscopy (SEM) and Transmission Electron Microscope (TEM). Antibacterial activity against *Staphylococcus aureus* was evaluated, confirming antimicrobial action of the encapsulated silver nanoparticles into the hydrogels.

## 1. Introduction

The application of nanomaterials, generally ranging from 1 to 100 nanometers (nm), is an emerging area of nanoscience and nanotechnology. Nanomaterials may offer answers to technological challenges in diverse fields, such as catalysis, solar energy conversion, water treatment, and medicine. These kinds of materials often show exclusive and considerably changed chemical, physical, and biological properties [1,2,3]. The intrinsic properties of metallic nanoparticles depend mainly on their size and shape features [4]. Copper, gold, and silver have been utilized frequently for the preparation of stable dispersions of nanoparticles to a large range of applications in several fields, including medical and food applications, among others [1,5,6]. In this context, silver nanoparticles (AgNPs) play a key role in nanoscience, principally in nanomedicine. Among the broad variety of applications, the antibacterial activity is one most investigated due to their exclusive physical and chemical features [7,8,9]. Successful prevention and treatment of a large range of diseases triggered by microorganisms that include viruses, bacteria, fungi, and parasites are of tremendous significance to public health and a huge challenge to pharmaceutical science [1,10]. Several bacteria, such as *Escherichia coli*, *Pseudomonas aeruginosa*, and *Staphylococcus aureus*, among others, are the etiological agents of numerous infectious diseases [11].

Nanoparticle properties are influenced by factors that include the method of selection for nanoparticle synthesis and kind of stabilizer utilized [11]. Metal nanoparticles are prepared and stabilized via physical and chemical techniques. By the chemical route, the nanoparticles can be prepared through chemical reduction, electrochemical, and photochemical reduction methods. Previous reports have shown that the morphology, size, size distribution, stability, and features of the metal nanoparticles are powerfully governed by experimental parameters, the kinetics of interaction of metal ions with reducing agents, and the type of stabilizing agent that interacts with metal nanoparticles. Therefore, the development of novel synthesis strategies that allow control of the morphology, size, stability, and nanoparticle properties have drawn significant attention [1,12,13]. There are three methods of stabilizing the nanoparticles: electrostatic charge stabilization, steric stabilization, and a combination of both (electrosteric stabilization) [11]. To stabilize and control the nanoparticle structures, numerous types of stabilizers have been utilized, such as surfactants, polymers, dendrimers, and biomacromolecules [14,15,16,17]. Among the stabilizing polymers, the hydrogels (i.e., made of Poly(vinylpyrrolidone) (PVP), poly(acrylamide) (PAM)) have gained special attention as a promising template to formulate nanoparticles as a novel concept composite/hybrid material [17,18].

Hydrogels are 3D, hydrophilic polymeric networks that are capable of absorbing large amounts of water, biological fluids, or other kinds of molecules [19]. The main features in hydrogels are porosity and pore size, which can easily be tunable by modifying the crosslink density in their network [20]. On the other hand, the start materials selected here for the hydrogel preparation, including polyvinyl alcohol (PVA) as main chain and maleic acid (MAL) as crosslinker, are frequently utilized as biomaterials, since they have high biocompatibility and have been Food and Drug Administration-approved [20,21,22]. Nanoparticle-hydrogel composites display multi-functional and stimuli responsive features, making them perfect for “smart” materials, including antimicrobial gels, barriers, matrices, and drug delivery vehicles, among others [23]. In this context, we have chosen the PVA polymers to prepare hydrogels, given their inherent properties, such as being stabilizing agents, biocompatible, and a multipurpose class of materials suitable for a broad range of applications. On the other hand, the characteristic tunable pore size can be ideal for synthesizing AgNPs with specific size distribution. Therefore, we have achieved the design of a novel strategy to control the size of AgNPs through a simple methodology using a single crosslinker (MAL) and single type main chain (PVA).

## 2. Materials and Methods

### 2.1. Materials

PVA (30–60 KDa), MAL, HCl, NaHCO_3_, AgNO_3_, and NaBH_4_ were utilized for hydrogel synthesis. These reagents were supplied by Merck (Darmstadt, Germany). Reagents to prepare phosphate buffer saline (PBS) (pH 7.4) and acetate buffer (pH 3.0) were supplied by Sigma-Aldrich (St. Louis, MO, USA). These reagents were supplied by Merck (Darmstadt, Germany). All solutions were prepared utilizing MilliQ water. *Staphylococcus aureus* ATCC® 25923, Brain Heart Infusion (BHI) agar, Luria-Bertani (LB), and peptone water were purchased in Merck (Darmstadt, Germany). Distilled water was utilized for the preparation of any solutions in antibacterial study.

### 2.2. Experimental Section

#### 2.2.1. Synthesis of PVA-MAL Hydrogels (PMALH)

Three types of PMALH formulations with different crosslinking degrees were prepared. The preparation of PMALHs was performed through the esterification of PVA with MAL—a dicarbolxylic acid molecule—according to Valdés et al. [24]. Concisely, the chemical syntheses were carried out by mixing a 10 wt % aqueous solution of PVA with an aqueous solution of MAL in the presence of 1 × 10^−1^ mol·L^−1^ HCl (pH 1). The final concentrations of MAL were 10, 20, and 30 wt % for PMALH-10, PMALH-20, and PMALH-30 formulations, respectively. The reaction mixtures were kept under reflux at 90 °C in a necked flask with magnetic stirring for 3 hours. After that, the reaction mixtures were put in an oven at 70 °C for another 3 hours to remove water by evaporation and increase the crosslinking degree. Next, the PMALH hydrogels were washed three times with NaHCO_3_ and one time with distilled water to remove the excess acid. Finally, the formulations were freeze-dried to achieve the xerogels. 

#### 2.2.2. Preparation of Hydrogel/Silver Nanohybrids

The three synthesized hydrogels were utilized to obtain hydrogel/silver nanohybrids. Briefly, a piece of each hydrogel of 200 mg was immersed into a solution of AgNO_3_ (50 mM) for 18 h and constantly stirred until reaching maximum absorption of the solution. Then, each swelled hydrogel piece was washed several times in order to remove any remnant of AgNO_3_ on the surface. After that, the hydrogels were transferred to another flask that contained the reducing agent NaBH_4_. The reaction was stirred for 30 minutes to reduce the Ag^+^ ions in order to produce the AgNPs. Finally, the nanoparticle-containing hydrogels were washed three times with distilled water and freeze-dried

#### 2.2.3. Swelling Studies

The water uptake process was calculated by equilibrium swelling ratio (%ESR) at desired time intervals. Briefly, dried hydrogel pieces (0.4–0.5 mm thickness, 1 cm^3^) were placed to swell in PBS (pH 7.4; 0.1 M) and acetate buffer (pH 3.0; 0.1 M) at room temperature. The swollen gel pieces were taken out from the swelling medium at regular time intervals, between 0 and 21 hours, and superficially dried with absorbent paper. Then, the hydrogel pieces were weighed and placed in the same solution. The measures were continued until a constant weight was reached (Equation (1):(1)$$\%\mathrm{ESR}=\frac{M_{s}-M_{d}}{M_{s}}\times 100$$
where, %ESR is the swelling index, *M*_s_ and *M*_d_ are the mass of the swollen hydrogel and the dried hydrogel or xerogel, respectively. To investigate the rate of water absorption, the water-intake process was calculated through the determination of the swelling index of the hydrogel at desired time intervals, as previously described. 

#### 2.2.4. Infrared Spectroscopy

Fourier-Transform Infrared (FT-IR) spectra of PMALH10, PMALH20, and PMALH30 were recorded on a Nicolet Nexus 470 spectrometer within the 4000–400 cm^−1^ spectral intervals. All spectra were obtained in KBr pellets from an average of 32 scans with 4 cm^−1^ resolution.

#### 2.2.5. Thermogravimetric Analysis (TGA)

TGA was performed utilizing a TGA analyzer Q500 (TA Instruments, New Castle, DA, USA). The samples were ramped from 60 to 600 °C at 10 °C·min^−1^ with a gas flow rate at 60 mL·min^−1^ in synthetic air and nitrogen atmosphere. The mass remaining was recorded throughout the experiment. The derivative thermogram analysis (DTG) was calculated using OriginPro software (OriginLab, OriginPro 8.5, Northampton, MA, USA).

#### 2.2.6. Scanning Electron Microscopy (SEM) Analysis

The hydrogel samples were cut and loaded in the copper stub. Next, the samples were stained with 0.7% (*w*/*v*) phosphotungstic acid, washed, and air-dried. The samples were observed in SEM mode in a Low-Voltage Electron Microscope (at a nominal operating voltage of 5 kV) LVEM5 (Delong Instruments, s.r.o., Brno, Czech Republic). The weight percentage of silver was acquired by Energy Dispersive Spectroscopy (EDS) (OXFORD INSTRUMENTS INCAx-SIGHT, year 2005), coupled to a SEM microscope.

#### 2.2.7. Transmission Electron Microscopy (TEM) Analysis

In vitro generated AgNPs by hydrogel were observed using a Hitachi Transmission Electron Microscope HT7700 (Tokyo, Japon). The AgNPs were released from the hydrogels by hydration with three volumes of distilled water and sonicated in a bath at an approximate frequency of 20 kHz for 5 minutes. Later, each sample was prepared by placing a drop of nanostructure solution on a Lacey Carbon TEM Grid. Microscopic analysis was carried out at the Center for the Development of Nanoscience and Nanotechnology—CEDENNA, USACH.

#### 2.2.8. Antibacterial Activity

##### Screening of Antimicrobial Activity of PMALHs Against Staphylococcus Aureus

To evaluate the antibacterial activity of synthesized hydrogel/silver nanohybrids, Gram-positive strain *S. aureus* ATCC® 25923 was used as a model pathogen. For this, disks of 6 mm diameter of PMALH hydrogels were cut and then were placed on BHI agar plates; the inoculum (100 µL) containing *S. aureus* in the range of 1.0 × 10^6^ CFU·mL^−1^ was spread previously on the agar surface. Then, the plates were incubated at 37 °C for 24 h. Chloramphenicol (CHL) discs of 30 µg as a positive control were used and hydrogels without AgNPs as a negative control were used. In previous tests, negative control resulted with negative activity on *S. Aureus* (data not shown). All screening tests were performed in duplicate. Antibacterial activity was calculated by the formation of the inhibitory zone surrounding film disks against the microorganism. 

##### Quantitative Test of Antibacterial Activity of PMALHs Against Staphylococcus Aureus

For this assay, *S. aureus* was incubated in 1 mL of LB broth at 37 °C until the turbidity equivalent reached a 0.5 McFarland standard. After, 100 and 200 mg of each hydrogel/silver nanohybrid was put in contact with the bacteria and then incubated for 24 hours at 37 °C. After that each culture was tested; serial dilutions were made in 0.1% sterile peptone water. From each of these dilutions, 100 microliters were obtained, which were planted on a plate count agar and incubated at 37 °C for 24 h. Subsequently, viable cell counts were performed. All trials were performed in triplicate. For this assay, medium alone as a negative control was used and chloramphenicol as a positive control at a dose of 30 µg·mL^−1^ was used.

## 3. Results and Discussions

### 3.1. Preparation of PMALHs

The synthesis of PMALHs was performed as is depicted in Figure 1. Succinctly, the PVA chains were crosslinked by esterification reaction using HCl as a catalyst. Specifically, the hydroxyl groups of PVA were successfully esterified with carboxyl groups from MAL. The characterization study from %ESR, TGA, and FT-IR confirmed the covalent ester bond between PVA and MAL to form the formulation that behaves as a hydrogel at 25 °C. Crosslinking degrees of 10%, 20%, and 30% *w*/*w* of MAL were used, given that the hydrogel can present a significant porosity difference. This feature is key to the subsequent synthesis of AgNPs.

### 3.2. Preparation of Hydrogel/Silver Nanohybrids

The PMALH synthesized hydrogels were used as template to obtain nanohybrids based on a silver nanoparticle-hydrogel. In Figure 2, a scheme of the synthesis process for the preparation of these nanohybrids is illustrated. During the hydrogel swelling in the silver nitrate solution, the silver ions (Ag^+^) are stabilized into the hydrogel pores. The metal ions are fixed to the functional groups of hydrogel networks (alcohol “–OH” and carboxylic groups “–COO^−^“ available). Therefore, large amounts of silver ions can be trapped in the cavities of the hydrogel networks. Subsequently, at the intermolecular level, in situ reduction of silver ions to nanoparticles (using NaBH_4_) is occurring and AgNPs are stabilizing into PMALH hydrogel networks. The initial appearance of the hydrogel (colorless) changes quickly to dark brown given the in situ reduction process of silver. Unlike the hydrogel, the supernatant maintains the same appearance (colorless). In agreement with previous works, the produced nanoparticles in the network can inhibit the aggregation for an extended period of time and they can easily be desorbed toward the water [17].

### 3.3. ESR Results

This evaluation is very important since it allows us to confirm that each prepared hydrogel has a unique crosslinking degree with a specific pore size, which will play a key role during AgNPs preparation, where each formulation will be utilized as a size template pattern for the preparation of AgNPs. Therefore, the trials were performed with the goal of evaluating the swelling ability of the synthetized hydrogels utilizing diverse crosslinking degrees (PMALH10, PMALH20, and PMALH30) and two different pH values (pH 7.4 and 3.0) at 25 °C.

Figure 3 displays the swelling index for the three hydrogels. The graphics of %ESR show an increase in the swelling index throughout the time for all PMALHs. For the set of PMALHs, the index swelling in the first section increased quickly and then bit by bit (after 5 h). This behavior is owing to the formulations attaining a maximum swelling that is constant throughout the time. The PMALH10, PMALH20, and PMALH30 reached the swelling equilibrium at about 5 h (see Figure 3). 

The swelling index depends on the polymer characteristics, the average molecular weight, flexibility of the polymer chain, the crosslinking degree, and the network mesh size, among others. Moreover, the swelling index depends on external conditions, such as temperature and pH [25]. Regarding the pH, the prepared hydrogels presented pH-dependent swelling behavior owing to ionic networks. In this context, the three hydrogels absorbed a higher amount of water at pH 7.4 than pH 3.0. The ionic networks contain acidic pendant groups provided by the crosslinker (MAL), which have two types of pKa (pHa1 = 1.93 and pKa2 = 6.59) [26]. As has been demonstrated in previous articles, this characteristic at a certain pH provides higher ionization degree in the matrix, causing an increase of electrostatic repulsion between chains from the networks [27]. Therefore, the electrostatic repulsion generated a higher uptake of solvent into the matrix increasing the size of the hydrogel [28]. Furthermore, at all pHs a higher swelling ratio was observed with PMALH10 (Figure 3A); this is due to the lower crosslinking density and higher flexibility of the hydrogel. Specifically, at pH 7.4 and 3.0, PMALH10 presented an ESR with values near to 600% and around 500%, respectively. On the contrary, the PMALH20 and PMALH30 showed a significant ESR reduction, (PMALH20 at pH 7.4 = 400% and pH 3.0= 350%, PMALH30 at pH 7.4 = 300% and pH 3.0= 250%, as represented in Figure 3B,C). Furthermore, the swelling degree detected might be associated with the absorption process that governs the diffusion of water molecules and other hydrophilic molecules, such as salts, into the pores of the hydrogel network [29].

### 3.4. Thermogravimetric Analysis Results

The PMALH analysis and characterization was studied through thermogravimetric analysis (TGA). Figure 4 displays the TGA curves from each of the three different synthetized PMALHs with different concentrations of MAL. Moreover, the thermal stability was measured, where it is possible to divide the thermogram in four regions. The first region occurred between 40 and 180 °C, mainly owing to the loss of moisture and physically weak and chemically strongly bound water [27]. This is possible due to the free PVA concentration; the mass loss of three formulations was very similar (90.3%, 87.8%, and 88.4%). However, in PMALH10 it was observed that the water was physically lower than the water chemically because there were more free chain PVAs. The second region occurred between 180 and 280 °C, attributed to the degradation of free PVA. The third region between 280 and 385 °C was attributed to the formed hydrogel, mainly between PVA and MAL. The fourth region occurred between 325 and 450 °C with a crosslinker between the chain of PVA and MAL. A final decomposition of carbonaceous matter at 485 °C was observed.

### 3.5. DTG Curves and Deconvolution Analysis

In the analysis of DTG (Figure 5), different maximum temperatures of degradation was observed. The first signal (near to 350 °C) could correspond to the covalent ester bonds formed between PVA chains and MAL (partially bound at one end with PVA), as depicted Figure 5b. The second signal (neat to 425 °C) may be attributed to the covalent ester bonds PVA-MAL-PVA (bound at the two ends to PVA chains), as depicted Figure 5c. On the other hand, the different areas of maximum temperature of degradation can be attributed to the different crosslinking degrees of each prepared formulation.

### 3.6. FT-IR Results

Figure 6 showed the spectrum of FT-IR with some characteristic signals that demonstrated the formation of the hydrogel. Between 3050 and 3700 cm^−1^ there is a strong signal from strength vibration that may be attributed to the alcohol of the PVA chain and hydration grade of the hydrogel. At 2920 cm^−1^, the characteristic signal of alkyl (R–CH_2_) stretching modes of the PVA chain was observed [27]. Moreover, typical signals of the PVA spectrum stretching band at 2840 cm^−1^ were observed. The crosslinking formation between PVA and MAL can be observed in the signal at 1640 cm^−1^, which corresponds to the covalent ester bonds formed between the alcohol of PVA chains and carboxylic acid of MAL [30].

### 3.7. SEM and TEM Analysis

SEM analysis proves the morphological differences in both stages of the hydrogel. The three types of PMALHs without AgNPs and PMALHs with AgNPs are illustrated in Figure 7. Specifically, Figure 7A–C shows SEM micrographs of hydrogels without AgNPs (PMALH10, PMALH20, PMALH30). SEM micrographs of hydrogels with AgNPs ((PMALH10-AgNPs, PMALH20-AgNPs, PMALH30-AgNPs) are shown in the Figure 7D–F. On the SEM micrograph of PMALH10 and PMALH20, the smooth walls and an assembly of marked fiber networks can be observed. However, the SEM micrograph of PMALH30 shows a saturated surface, which makes it difficult to identify fibers from the network. These types of assembly with suitable permeability play a key role to allow the Ag^+^ ions diffusion through the pores when they were absorbed. The three systems presented irregular edges with well-marked reliefs on the surface. The common characteristic in the series of hydrogels was the presence of valleys between the reliefs that gradually decreased in size from PMALH10 to PMALH30.

A change can be observed in the hydrogels morphology after AgNPs stabilization. SEM images of hydrogel/silver nanohybrids show morphology like flowers; the AgNPs within the hydrogel network caused an increased porous structure in the nanohybrids (Figure 7D–F). The elemental composition of hydrogel/silver nanohybrids revealed that the weight percentages of silver given by Energy Dispersive X-Ray Analysis (EDX) were about 1.68% to 3.07%.

TEM images (Figure 8A–C) show the AgNPs stabilized within the hydrogel network synthesized with different crosslinking degrees (hydrogel/silver nanohybrids). From the micrographs, 170 nanoparticles were analyzed and their size distribution were calculated, as is illustrated in the respective histogram. Gaussian curves were fitted to the distributions; the curve center corresponds to the average nanoparticle diameter. For the samples of PMALH10-AgNPs, PMALH20-AgNPss and PMALH30-AgNPs, the average AgNP sizes were 6.48 ± 0.07 nm, 4.80 ± 0.20 nm, and 3.07 ± 0.07 nm, respectively, with a spherical shape and narrow size distribution. According to the AgNP size and their distribution within the hydrogel network observed by TEM images, it is possible to confirm that the available free-network spaces or cavities in the hydrogel networks allow growth and stabilization of the nanoparticles with a narrow size distribution. With these results it can be confirmed that the PMALH hydrogel network can be used as a template to prepare tunable-size AgNPs, such as those illustrated in Figure 8.

### 3.8. Evaluation of Antibacterial Activity

Recent studies suggest that the antibacterial activity of AgNPs predominantly depends on the size of the particles (the smaller they are, the better, i.e., 1–10 nm), which have a direct interaction with the bacteria [17,30,31]. Consistent with the above, the results of this article indicate that hydrogel/silver nanohybrids have a significantly stronger antibacterial effect as the AgNPs size decreases. Figure 9 displays the screening of antimicrobial activity of PMALHs against *Staphylococcus aureus*. According to these results, the antibacterial activity occurs mostly owing to the release of AgNPs from the PMALHs. The inhibition area follows in an increasing order: PMALH10 (0.8 cm) < PMALH20 (1.0 cm) < PMALH30 (1.5 cm). This conduct is expected due to smaller AgNPs, as in PMALH30 (∼3 nm), which are responsible for killing a greater number of *S. aureus* colonies, as is demonstrated in Table 1. Moreover, in the same table two doses for each formulation were evaluated, demonstrating that the concentration improves the activity.

## 4. Conclusions

We have developed a versatile method of synthesis in situ of AgNPs with a well-defined size using hydrogels with different crosslinking degrees as a template. The cross-linking degree of hydrogel networks has regulated and stabilized the nanoparticle size; that is to say, the distribution sizes of nanoparticles depend on the crosslinking degree of the hydrogel network. Therefore, we have achieved the design of a novel strategy to control the size of AgNPs through a simple methodology using a single crosslinker (MAL) and single-type main chain (PVA). TEM images have demonstrated the well-defined size of nanoparticles obtained and extracted from hydrogel networks. Concerning hydrogel characterization, the results of FT-IR, TGA, and SEM exhibited that all PMALH hydrogels were successfully prepared. Moreover, the hydrogels have good swelling property. The swelling assays revealed that the prepared hydrogels are pH-sensitive. The antibacterial activity evaluated in this study is consistent with previous articles that demonstrate that the AgNPs have antibacterial activity, which in this case was against the *S. aureus*. Moreover, it is concluded that the antibacterial activity against *S. aureus* depends on nanoparticle size; namely, the antibacterial effect is better as the nanoparticle size decreases.

## Figures and Tables

**Figure 1 polymers-11-00716-f001:**
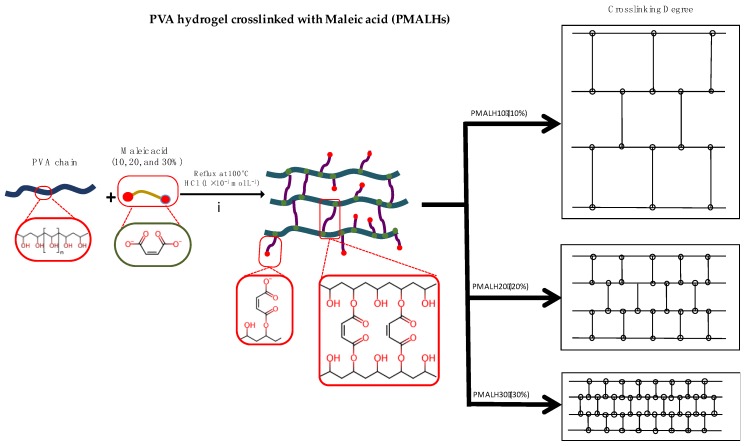
Scheme of synthesis of PMALH Hydrogels. Esterification reaction of PVA with MAL at three different ratios.

**Figure 2 polymers-11-00716-f002:**
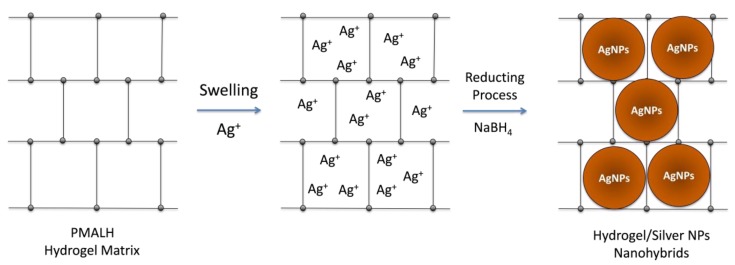
Preparation scheme of hydrogel/silver nanohybrids (PMALH-AgNPs).

**Figure 3 polymers-11-00716-f003:**
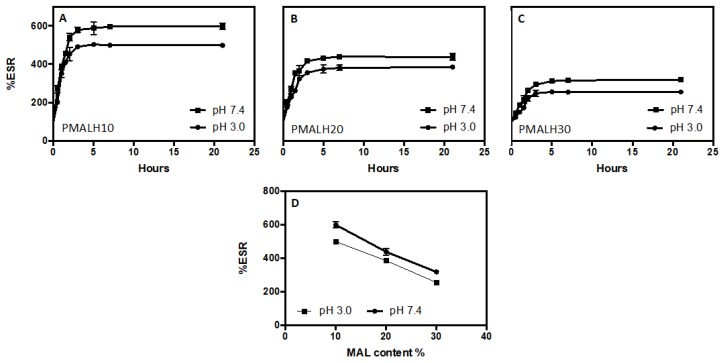
Swelling index of (**a**) PMALH10, (**b**) PMALH20, and (**c**) PMALH30 in two different buffers (pH 3.0, 7.4) with respect to time. (**d**) %ESR of PMALH10, PMALH20, and PMALH30 at pH 7.4 and 3.0 with respect to time.

**Figure 4 polymers-11-00716-f004:**
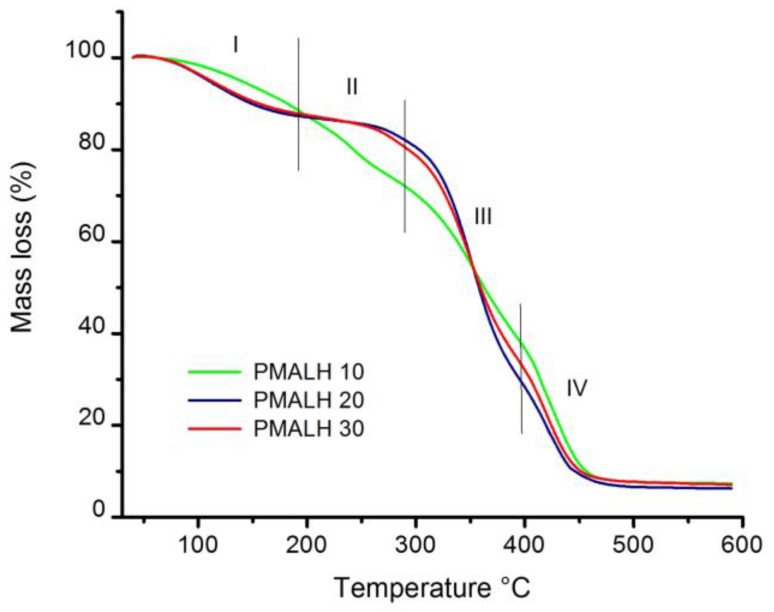
TGA (Thermogravimetric) thermogram for polyvinilalcohol (PVA) crosslinker with MAL.

**Figure 5 polymers-11-00716-f005:**
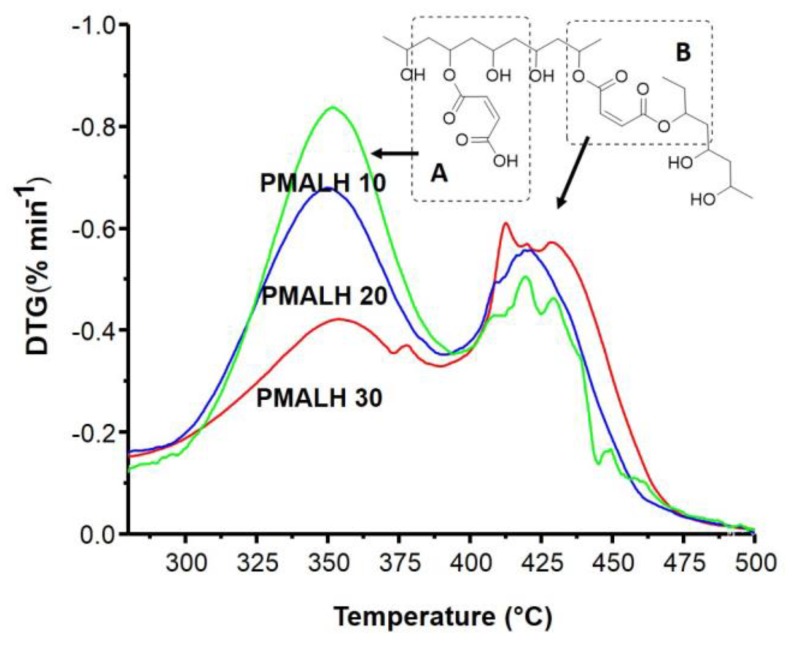
DTG between 280 and 500 °C and possible crosslinking. (**A**) fraction PVA-AM and (**B**) fraction PVA-AM-PVA.

**Figure 6 polymers-11-00716-f006:**
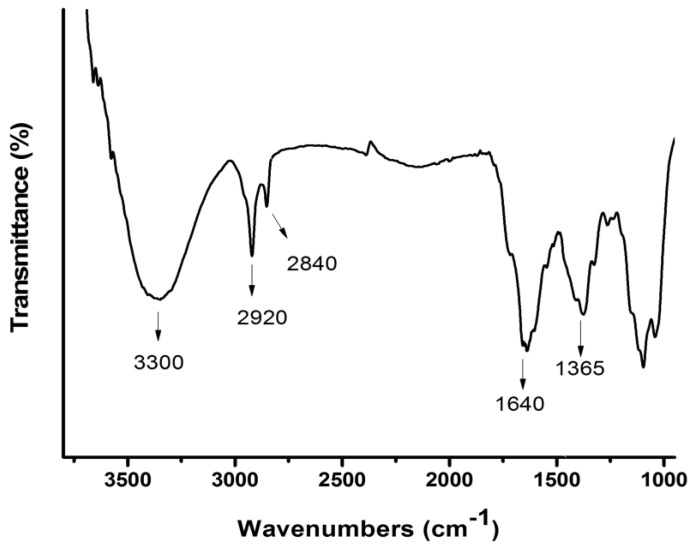
Typical FT-IR spectra of PMALHs.

**Figure 7 polymers-11-00716-f007:**
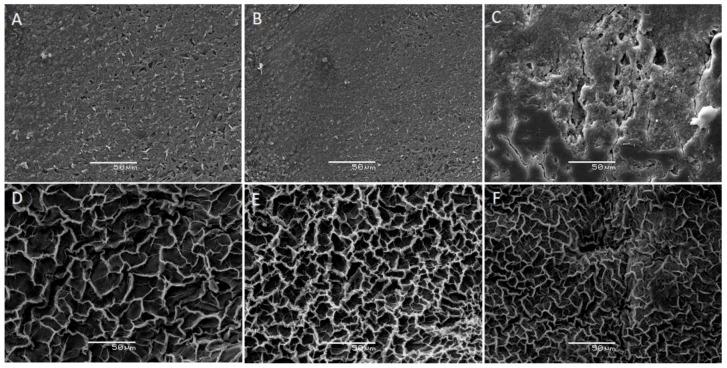
SEM images of (**A**) PMALH10, (**B**) PMALH20, (**C**) PMALH30 hydrogels, (**D**) PMALH10-AgNPs, (**E**) PMALH20-AgNPs, and (F) PMALH30-AgNPs nanohybrids.

**Figure 8 polymers-11-00716-f008:**
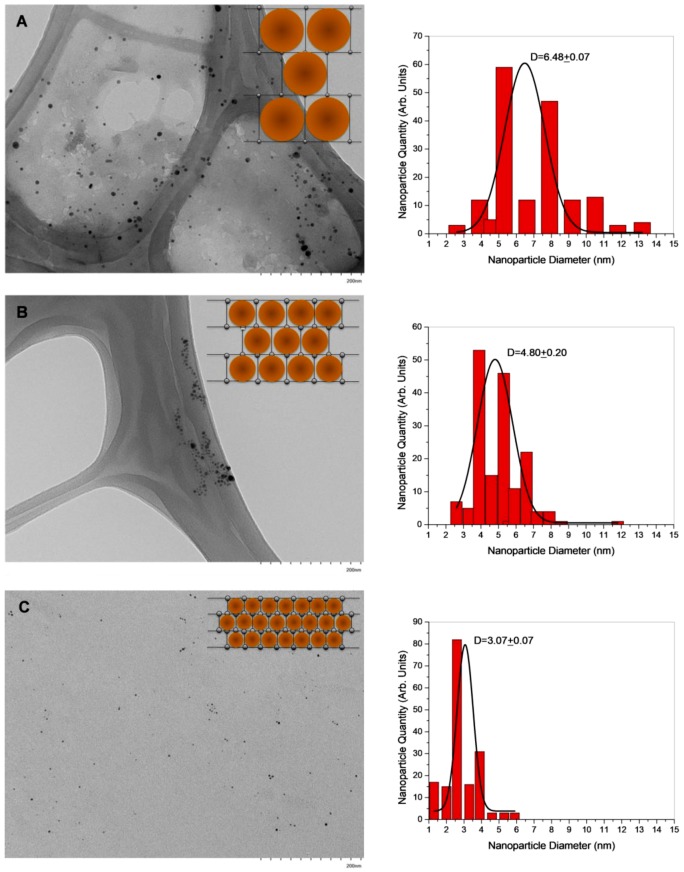
TEM images of (**A**) PMALH10-AgNPs, (**B**) PMALH20-AgNPs, and (**C**) PMALH30-AgNPs nanohybrids and their respective histogram.

**Figure 9 polymers-11-00716-f009:**
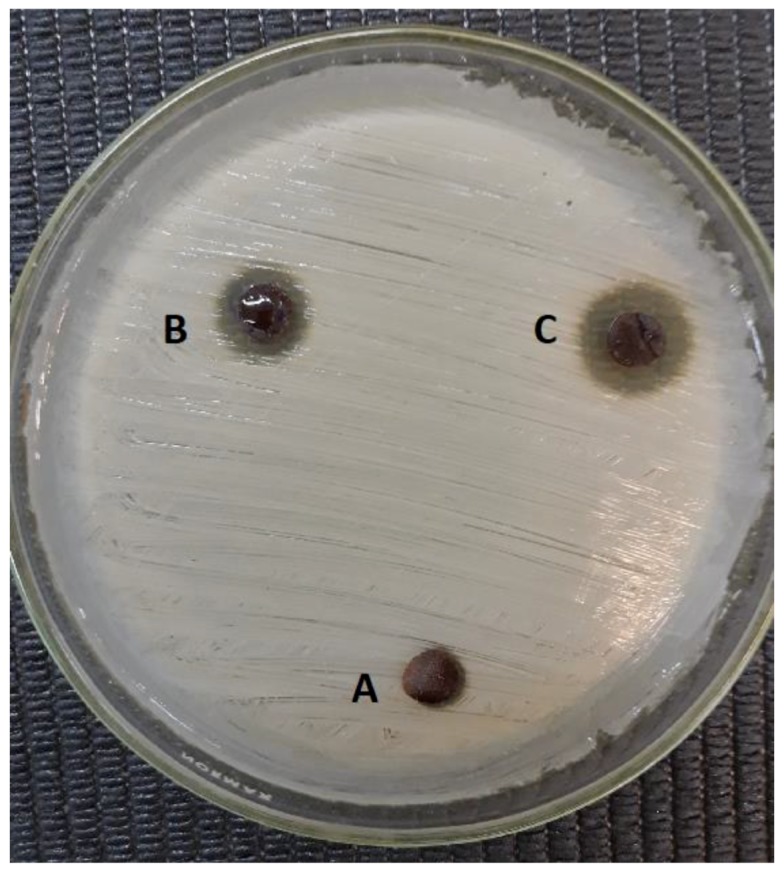
Screening of antibacterial effect of (**A**) PMALH10-AgNPs, (**B**) PMALH20-AgNPs, and (**C**) PMALH30-AgNPs nanohybrids against *S. aureus*.

**Table 1 polymers-11-00716-t001:** Quantitative assay of antimicrobial activity of PMALH on *S. aureus*.

Assay	Hydrogel Doses (mg)	*S. aureus* (UFC·mL^−1^)
*S. aureus*	-	23 × 10^8^
*S. aureus* + PMALH10	100	13 × 10^5^
	200	11 × 10^5^
*S. aureus* + PMALH20	100	12 × 10^5^
	200	93 × 10^4^
*S. aureus* + PMALH30	100	76 × 10^4^
	200	30 × 10^4^
*S. aureus* + CHL (Positive control) Medium alone (Negative control *)	- -	- -

* Corresponding to the culture medium without bacteria or hydrogel.

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
