# Peer review of "Preparation of Hydrogel/Silver Nanohybrids Mediated by Tunable-Size Silver Nanoparticles for Potential Antibacterial Applications"

_polymers, 2019, doi:10.3390/polym11040716_

Round 1

Reviewer 1 Report

This manuscript described the preparation and characterization of a composite material of PVA hydrogel/AgNPs. It should be improved according to the comments below before acceptance.

Line 65. Why are the hydrogels made of synthetic polymers considered as biomacromolecules?

Line 176. "by polymerization by esterification" is confusing.

Line 233-234. The expression is not scientific.

Line 263-264. "the water physically" and "the water chemically" is not a correct expression.

Figure 5. What is DTG? Please give the full name at its first appearance. Please also introduce it in the Methods part.

Line 280. How does Figure 5A show the"greater degree of crosslinking"?

Line 319 and Line 321. "Figure 4" should be "Figure 7".

Figure 7. "Porous structure" is not clear from the given SEM images.

Author Response

Dear Reviewer 1.

Please find attached the response document.

The Best 

Esteban Durán 

Reviewer 2 Report

This work was to prepare hydrogel/silver nanohybrids for the antibacterial application. The authors concluded that the antibacterial activity against S. aureus depends on nanoparticle size. Several questions are listed.

1.          The authors must describe clearly the aims (unmet medical needs) for developing this hydrogel/silver nanohybrids used for antibacterial purposed. The antibacterial effects of shape/ size of silver nanoparticles have been discussed [RSC Adv., 2014, 4, 3974–3983. Appl Environ Microbiol. 2007 Mar; 73(6): 1712–1720.]. In addition, the strategy of controlled silver/gold nanoparticles synthesis in the semi-hydrogel networks has also been proposed [Materials Chemistry and Physics 103 (2007) 278–282. Carbohydrate Polymers 75 (2009) 463–471. Biomater. Sci., 2014, 2, 257.]. Furthermore, hydrogel itself may also contribute to the antibacterial effects [Soft Matter, 2011, 7, 8725.], so an additional group of a hydrogel is necessary for the comparison of hydrogel/silver nanohybrids. What is the novelty of the current study?

2.          Section 2.2.8.2. A positive control group is necessary.

3.          Line 178 & line 206. The authors must define ESR.

4.          Section 3.1. How did the authors confirm the produced PMALH is the same as or similar to the results of reference [24]. For example, did the authors check the average molecular weight?

5.          Section 3.8. For the valuation of antibacterial activity, the study design is too simple. What is medicine used for the positive control group? How can the authors know the antibacterial activity contributed by AgNP or hydrogel?

6.          Table 1. The sample size (N=?) must be included. Statistical analysis is necessary.

7.          The font for Staphylococcus aureus or S. aureus should be Italic.

8.          Overall, the authors have to clearly state the new findings or the purposed for application fields to strengthen the novelty of the article. 

Author Response

Dear Reviewer 2,

Please, find attached the document with responses to your suggestions.

Best Regards,

Esteban Durán 

Round 2

Reviewer 2 Report

Table 1. The author should clarify the definition of a negative control group.

Author Response

Dear reviewer, 

Please find enclosed the point-by-point response to the reviewer’s comments

Sincerely 

Esteban Durán 
